# Optimization of Cellulosic Fiber Extraction from Parsley Stalks and Utilization as Filler in Composite Biobased Films

**DOI:** 10.3390/foods11233932

**Published:** 2022-12-05

**Authors:** Hulya Cakmak, Matthijs Dekker

**Affiliations:** 1Department of Food Engineering, Hitit University, 19030 Corum, Turkey; 2Food Quality and Design Group, Department of Agrotechnology and Food Sciences, Wageningen University, 9 6708WG Wageningen, The Netherlands

**Keywords:** parsley stalk, waste, optimization, cellulose fiber, biodegradable film

## Abstract

Food waste is an abundant source of cellulose which can be extracted via mild alkali treatment. The extraction conditions of cellulose fibers can be optimized for reduced chemical and energy use and optimal functionality. This study focused on the optimization of alkali extraction of lignocellulosic fiber from parsley stalks by building an experimental design with the response surface method with alkali concentration (2, 6, and 10%, *w*/*v*), fiber:alkali ratio (0.02, 0.035, and 0.05; *w*/*v*) and extraction temperature (40, 70, and 100 °C) as independent variables, in order to evaluate the effects of extraction conditions on fiber yield and composition of parsley stalks extract (PSE). Following the optimization, PSE and untreated fibers (PF) were incorporated as filler into gum Arabic–sodium alginate-based films, and film properties such as water vapor permeability, optical and thermal properties, Fourier transform infrared spectra and surface morphology of the films were analyzed for evaluating the compatibility of these fillers with the composite film matrix. The optimal extraction conditions were determined as 2% alkali, sample:alkali ratio of 0.0276 and extraction temperature of 40 °C. PSE extracted at optimal conditions was added to the composite films, and water vapor permeability and optical properties were improved by up to 10% PSE compared to films with PF.

## 1. Introduction

Agricultural and food wastes are rich in lignocellulosic materials as they are composed of leaves, peels, seeds, pomace, husk, stalks or damaged crops. Those wastes are estimated at nearly 1.3 billion tons per year globally [1,2,3]. Since these materials are extracted from biomaterials, they are cheap, renewable, biodegradable and abundant. They can be used in several applications due to their good mechanical properties e.g., for food packaging, textile or paper [4,5,6].

Lignocellulosic wastes are a good source of natural fibers that are mainly composed of cellulose, hemicellulose, lignin and extractables. Alkaline pretreatment of these wastes helps to increase the cellulose content through the degradation of lignin and hemicellulose and their removal from the liquor by consecutive washing steps [4,6,7,8,9,10]. Moreover, alkaline treatment results in surface modification, increased fiber porosity, enhanced digestion of lignocellulose, improved tensile properties and increased compatibility of the extracted fiber with packaging film matrices resulting in improvements of mechanical and optical film properties [4,7,11]. There are various alkaline pretreatment conditions described in the literature regarding the extraction period, extraction temperature, alkaline concentration and sample/alkaline ratio. Simultaneous optimization of these conditions has not been studied yet. As stated by Manna et al. [12], used sample/solvent ratios range between 1:50 and 1:10, whereas alkaline concentrations range generally between 0.8–15%, although up to 40% alkaline concentration usage has been found in the literature [10,11,13,14]. Reported extraction times vary from 0.5 to 48 h; however, extraction times between 2–8 h are most commonly used [11,12,15]. In order to accelerate the extraction process and the interaction of fiber with solvent, elevated extraction temperatures (80–100 °C) instead of extraction at room temperature are employed [10,11,13,16]. Optimization of the lignocellulosic material extraction conditions is important, not only to increase the yield and quality of fibers but also for the effective utilization of agro-food waste materials or by-products in accordance with sustainable production practices [10]. 

Recent studies have shown that waste or by-products from agricultural or food industries such as apple pomace [16], artichoke leaf [11], palm leaf stalk [17], parsley [9,18], sugarcane bagasse [14] and corn and rice husks [15], can be used as a source of cellulosic fractions and can be utilized for the production of micro and nano-sized fibers. Besides, cacao pods, rice husks, and parsley and spinach stems can be converted into bioplastics following the digestion of lignocellulosic material with trifluoroacetic acid [19]. Among them, parsley stalks are a good source of cellulosic material, although they are discarded as inedible food waste following harvesting due to their tough and fibrous texture [9,18]. Even though there have been a few studies of cellulosic material extraction from parsley stems or wastes by chemical treatments, there is a lack of knowledge about how alkali extraction conditions may affect the yield and composition of the extracted fibers.

The environmental burden of petroleum-based plastic packaging disposal in the environment should be immediately tackled [20,21]. In addition, the efficient recycling or composting of petroleum-based plastics is rather complex. Consequently, utilization of biobased and biodegradable bioplastics from natural biomasses such as proteins and polysaccharides (starch, gums, chitosan, alginate, cellulose, etc.) or bioplastics such as polylactic acid (PLA), polyhydroxyalkanoates (PHA) or polybutylene succinate (PBS) is exponentially increasing as replacements for petroleum-based plastics [22,23]. Gum Arabic is a low-cost and biocompatible material that is extracted from acacia trees [24,25,26]. It has a negatively charged protein–polysaccharide complex biopolymeric structure and can be used in food packaging film production. It is generally used in a composite form with the addition of different polysaccharides such as alginate, cellulose, chitosan or proteins for improvement of its poor barrier and mechanical properties [24,25,26,27,28]. 

The objectives of this study were (1) the optimization of alkali extraction conditions from parsley stalks as a lignocellulosic fiber source, and (2) the utilization of fiber extracted at optimal conditions in gum Arabic–sodium alginate composite films as fillers for improving the chemical compatibility, optical and thermal properties of the biobased films in comparison with untreated fiber.

## 2. Materials and Methods

### 2.1. Materials

Fresh parsley stalks were collected from the local bazaar in Corum, Turkey. Gum Arabic (Merck, Germany), Na-alginate (99%, Protanal PH 6160, FMC International, Willingtown, Ireland), glycerin (99.5%, Smart Kimya Ltd., Sti., Izmir, Turkey), NaOH (≥99%, Tekkim Kimya San. Tic. Ltd., Sti., Bursa, Turkey), H_2_SO_4_ (95–98%, Isolab, Eschau, Germany), HCl (37%, Isolab, Eschau, Germany), magnesium nitrate hexahydrate (99%, Merck, Darmstadt, Germany) were supplied by local distributors.

### 2.2. Production of Alkali Treated Parsley Stalks Fiber Extract (PSE) According to Response Surface Design

Fresh parsley stalks were removed from leaves, washed with tap water and excess water was drained. The stalks were dried at room temperature for 5 days, and finish dried at 50 °C for 8 h in an oven (Memmert, UN 55, Schwabach, Germany), ground with hammer mill (Brabender, SM3, Duisburg, Germany) and sifted through a 355 µm sieve for producing the PF powder ready for alkali extraction. 

Alkali treatment was performed according to the following steps given in Figure 1, with the treatment conditions of the experimental design as given in Table 1 and following those given in the study of Sogut and Cakmak, with some modifications [29]. The experimental design was built by using the Minitab 15 statistical software and response surface method using Box–Behnken design employing three factors, namely, alkali concentration (2, 6, and 10%, *w*/*v*), fiber:alkali solvent ratio (0.02, 0.035, and 0.05; *w*/*v*) and extraction temperature (40, 70, and 100 °C) at three different levels in order to evaluate the effect of extraction conditions on the yield and properties of PSE. The extractions were performed for 1 h at each design point in a preconditioned water bath (Wisebath, WB22, Daihan Scientific Co., Ltd., Seoul, South Korea). Following the extraction, the pH of the fiber and alkali mixture was neutralized using dilute HCl and centrifuged at 4000 rpm for 6 min (Rotofix 32A Hettich Zentrifugen, Tuttlingen, Germany). The neutralized mixture was washed with distilled water and the washing/centrifugation cycle was repeated until the supernatant was clarified (Figure 1). The extracted fibers were then dried at 60 °C for 24 h in an oven (Memmert, UN55, Germany). Alkali extractions at each design point were performed as duplicates.

### 2.3. Physicochemical Properties of PF and PSE

The moisture content of the samples was determined by drying them in an oven (Memmert, UN 55, Germany) at 105 °C for 4 h.

The ash of dried parsley stalks (PF) and PSE samples was determined using a muffle furnace (Protherm, PLF110/6, Ankara, Turkey) at 525 °C according to the method of AOAC 940.26 [30]. The average of triplicate measurements was reported.

The fiber yield of alkali extraction treatments was determined by a method similar to that given in Sánchez-Safont et al. [13], according to the following equation:(1)$$Fiber\;yield\;\left(\%\right)=\frac{W_{PSE}}{W_{PF}}\times 100$$
here *W_PSE_* is the weight of extracted fiber, and *W_PF_* is initial weight of (untreated) fiber. The average of triplicate analysis was reported.

The hemicellulose, lignin and cellulose amounts in the fiber samples were determined according to the methods given in the study by Ayeni et al. [31] and Sogut and Cakmak [29] with some modifications. Briefly, 0.3 g of fiber was mixed with 45 mL of 0.5 N NaOH and boiled for 3.5 h in a water bath. Then the mixture was filtered and dried at 105 °C. The difference between the initial weight of the sample and dried fiber was recorded as the amount of hemicellulose (H). For lignin analysis, 0.3 g of the sample was reacted with 72% H_2_SO_4_ at 25 °C for 2 h. Then the mixture was diluted to 3% with distilled water and heated to 120 °C for 1 h. The mixture was kept at room temperature for 24 h and dried after filtering with several washing steps. The weight of the dried insoluble fraction was recorded as lignin amount (L). Cellulose amount (%) was calculated by subtracting H%, L% and ash % from 100.

The water absorption capacity of the fibers (PF and PSE at optimal conditions) was performed according to the ASTM D570 standard with some modifications [6]. The fiber sample was dried at 105 °C in an oven for 4 h, and 0.25 g of the dry fiber was weighted into 50 mL conical centrifuge tubes. The tubes were filled with 50 mL distilled water and the samples were hydrated for 24 h at room temperature. Swelled fibers were removed from water and air dried on a filter paper for a few minutes, and the weight was recorded. The percentage of water absorbed per g of dry fiber was calculated, and the average of at least triplicate measurements was reported.

### 2.4. Production of Biodegradable Films with PF and PSE

Gum Arabic–Na-alginate composite films were produced by the solvent casting method according to the study by Abdin et al. [26] with some modifications. As control film, 1% (*w*/*v*) gum Arabic (GA) and 0.5% (*w*/*v*) Na-alginate (NA) powder was mixed with distilled water for 2 h at 70 °C at 900 rpm with a magnetic stirrer (WiseStir MSH20A, Seoul, Korea). After the mixture was completely homogenized, 40% glycerol (*w*/*w*, on GA + NA basis) was mixed with the film forming solution for 1 h under the same conditions. An amount of 150 mL of the prepared mixture was poured into a Teflon plate (ID: 21 cm). To produce fiber-added films, a similar procedure was followed, and untreated PF and PSE extracted at optimal conditions were ground into a fine powder with a lab-scale hammer mill (Brabender, SM3, Germany) and fibers smaller than 125 µm were added into the film forming solutions at 1, 5 and 10% (*w*/*w*, on GA + NA basis) after the mixing with glycerol. The films were then dried at room temperature for at least 48 h and conditioned in similar conditions given in the WVP measurement prior to the analysis. All the film samples were produced in triplicate.

### 2.5. Characterization of the GA–NA Composite Films

#### 2.5.1. The Water Vapor Permeability (WVP)

The WVP of the films was measured using the ASTM E96-96 gravimetric method [32]. One side of the film was exposed to 100% RH, while the other side was exposed to 52.5 ± 0.4% RH using saturated magnesium nitrate. The weight of the films which were mounted hermetically on permeability cups was measured with an analytical balance with a sensitivity of 0.001 g (Precisa Gravimetrics, XB220A, Dietikon, Switzerland). The permeability cups were placed in a desiccator, and the mass of the cups was recorded periodically (every 2 h for at least 30 h) at 26.8 ± 1.1 °C. The slope of the mass change versus time (g/h) was obtained from the graph and the water vapor transmission rate (WVTR) was calculated by dividing the slope (g/h) by the film area (m^2^) open to the testing environment (g/h.m^2^). The WVP (g/kPa.h.m) was calculated according to the following equation: (2)$$WVP=\frac{WVTR\times d}{P\left(100-R_{e}\right)}\times 100$$
here, *d* is the thickness of the film (m), *P* is the water vapor pressure (kPa) at the temperature of the storage environment, *R_e_* is relative humidity inside the desiccator. The analysis was performed in triplicate.

The thickness of the films was measured with a caliper for at least five different randomly selected points of the films and their averages were reported.

#### 2.5.2. Optical Properties of the Films

The transparency (T, %) of the films was measured by calculating the transmittance percentage at 450 nm with a UV-vis spectrophotometer (Shimadzu, UV-1601, Tokyo, Japan).

The absorption spectrum of the film strips (1 cm × 3 cm) was recorded between 400–800 nm (Shimadzu, UV-1601, Tokyo, Japan) to determine the opacity. The opacity of each film was calculated by dividing the area under the curve (absorbance unit × nm) with the thickness of the film (AU nm/mm) [29].

#### 2.5.3. Color Measurement

The color of the fibers and films were measured using CIE L*a*b* color space with a spectrophotometer (Konica Minolta, CM3600D, Osaka, Japan). In this scale, L* represents brightness (0: black, 100: white), while +a*: redness, −a*: greenness, +b*: yellowness and −b*: blueness values. The standard white calibration plate was used as a background for the films, and the average of at least ten measurements for each film sample was presented.

#### 2.5.4. Thermogravimetric Analysis (TGA)

TGA was performed similar to the method of Sogut and Cakmak [29] using a TG/DTA (Perkin Elmer Pyris Diamond, Waltham, CA, USA) instrument. The measurements were conducted using nitrogen as purge gas at a 50 mL/min flow rate. An amount of 4–5 mg of dry film samples were heated between 25–1000 °C with a heating rate of 10 °C/min. 

#### 2.5.5. Fourier Transform Infrared (FTIR)

The Fourier transform infrared (FTIR) spectra of fibers (PF and PSE) and the biobased films were recorded using an FTIR spectrometer (Nicolet iS50, Thermo Scientific, Madison, WI, USA) equipped with a diamond ATR module. All the spectra were recorded with a resolution of 4 cm^−1^, from 700 cm^−1^ to 4000 cm^−1^, and the analysis was performed at 25 °C. The average of duplicate measurements was reported.

#### 2.5.6. SEM Analysis

The surface morphology films were examined with a scanning electron microscope (SEM) (Zeiss EVO LS10, Oberkochen, Germany) with an accelerating voltage of 10 kV in the low vacuum environment. The film samples were sputter-coated with gold prior to the analysis. A magnification level of 1000× and 3000× was used to scan each film sample to obtain the microstructural images.

### 2.6. Statistical Analysis

The results of the experimental design and optimization analysis were performed with Minitab software (Vers. 15, State College, PA, USA) with a 95% confidence level. The analysis of variance (ANOVA) and Tukey’s multiple comparison tests were employed to determine the significant differences between the film samples at a confidence level of 95% using SPSS software (Vers. 16, SPSS Inc., Chicago, IL, USA).

## 3. Results and Discussion

### 3.1. Physicochemical Properties of PF and Evaluation of the Experimental Design

The moisture content of untreated parsley fiber (PF) was 3.12 ± 0.49%, while the ash content was 9.97 ± 0.75% (on a dry basis). The hemicellulose, lignin and cellulose contents of PF were determined as 81.36 ± 3.51%, 6.26 ± 1.53% and 2.72 ± 0.04%, respectively. In order to evaluate the effects of alkali extraction conditions, percentage fiber yield, cellulose, cellulose/hemicellulose, cellulose/lignin and lignin amounts were selected as the responses of the experimental design, as shown in Table 2.

Following the alkali extraction, the hemicellulose and extractables were efficiently removed as expected due to the hydrolysis of hemicellulose and lignin depolymerization [6]. The ash content of the PSE ranged between 2.0 and 8.5% (dry basis). Lignin% was proportionally increased due to the reduction of hemicellulose, which was the main component in the lignocellulosic structure of the PF. These results are similar to the findings obtained for the alkali treatment of banyan tree roots, bitter vine leaf wastes and *Thespesia lampas* plant [6,7,33]. The average cellulose of the PSE was found to be between 17.6–55.9%, whereas the fiber yield was found to be between 19.7–49.7%. The extraction yields of the PSE were in line with the results of the fibers extracted from apple pomace, artichoke petal leaves, palm leaf stalk and *Thespesia lampas* plant [7,11,16,17]. The highest cellulose percentage was obtained in the experimental conditions which gave the lowest fiber yield (run 13) due to the highest removal of hemicellulose for these conditions (the highest temperature and NaOH concentration). 

The effects of the extraction parameters (alkali concentration, fiber:solvent ratio and temperature) individually and their interaction on the responses are shown in Table 3, Table 4 and Table 5. According to the regression analysis (Table 3), the fiber yield was significantly affected by the alkali concentration, the temperature and the square of the alkali concentration (*p* < 0.05). Therefore, these significant terms were used for building the equation representing the fiber yield, as shown in Equation (3). The ANOVA results confirmed that this model was significant at a 95% level (*p* < 0.0001), with an insignificant lack-of-fit value (*p* = 0.646). Overall, the prediction capacity of the model for estimating fiber yield was quite successful which was confirmed by the high adjusted correlation coefficient (Adj-R^2^), the low standard deviation (S) and the prediction error sum of squares (PRESS).
Fiber yield (%) = 72.35 − 3.57C − 0.33T + 0.17C^2^(3)

The cellulose percentage of the extracted fibers was modeled successfully (*p* < 0.0001, Adj-R^2^ > 0.986) (Table 4). The square of alkali concentration and temperature together with the interaction of alkali concentration with the sample:alkali ratio and alkali concentration with temperature were found to have significant effects (*p* < 0.05). As shown in Equation (4), the important terms were analyzed again for the prediction of the cellulose percent. The lack-of-fit was determined to be insignificant (*p* = 0.417).
Cellulose (%) = 83.63 − 8.46C − 181.25S − 1.50T + 55.17C × S + 0.05C × T + 0.30C^2^ + 0.01T^2^(4)

The results of regression analysis (Table 5) for cellulose/hemicellulose demonstrated that the alkali concentration, temperature, the square of temperature and the interaction of concentration with temperature had significant effects (*p* < 0.05) and their correlations are represented by Equation (5). However, the prediction capacity of this response was found a bit lower compared to the fiber yield and cellulose percentage (Adj-R^2^ = 0.86). In addition, the lack-of-fit for this response was determined as significant (*p* = 0.007).
Cellulose/hemicellulose = 2.51 − 0.16C − 0.07T + 0.003C × T+ 0.0005T^2^(5)

In addition to those given responses, cellulose/lignin and lignin percentage prediction models were evaluated by regression analysis and ANOVA results. Although the temperature, alkali concentration and their square terms had a significant effect on the prediction equations, the cellulose/lignin and lignin percentage prediction models had a lower prediction capacity than the models for fiber yield and cellulose percentage (Adj-R^2^ = 0.81 and 0.75, respectively). Therefore, the fiber yield, cellulose percentage and cellulose/hemicellulose parameters were selected for use in further optimization studies. We maximized those responses to optimize the NaOH concentration, sample:alkali ratio and temperature. The optimum conditions were found to be 2% alkali concentration, a sample:alkali ratio of 0.0276 and a 40 °C extraction temperature by the response optimizer with a composite desirability of 0.654. According to the proposed optimization conditions, the predicted responses were found to be 50.0% fiber yield, 28.8% cellulose and 0.64 cellulose/hemicellulose ratio. These optimization extraction conditions were confirmed by the experimental test, and the predicted responses were found to be similar to the experimental results (*p* > 0.05). So, the PSE extracted at the optimum conditions (C: 2%, S: 0.0276, and T: 40 °C) was used as a filler in the GA–NA composite films in further steps.

The water holding capacity of untreated PF (490 ± 35%) was found to be significantly lower than that of PSE extracted at the optimum conditions (1240 ± 120%). Alkaline treatments of lignocellulosic biomass can cause several changes in chemical and physical structure by disruption of the lignin barrier and increasing accessibility to cellulose, namely an increased swelling and surface area, while increasing the cellulose crystallinity and degree of polymerization [7,8]. In addition, it may improve the wetting properties of extracted fiber by exposing the reactive sites of cellulose and hemicellulose and thus increasing the affinity for water [6]. Similar to our findings, a more than twofold increase in the water absorption (or swelling) capacity was observed for bitter vine and artichoke petal leaves fiber with alkali treatment [6,11].

The color values of the PF were found to be L* = 63.95 ± 0.78, a* = −1.32 ± 0.23 and b* = 18.51 ± 0.62, while the color of the PSE extracted at the optimum conditions were found to be L* = 70.91 ± 0.51, a* = −2.73 ± 0.07 and b* = 15.95 ± 0.41. The noticeable increase in the L* value and reduction in b* of the PSE compared to PF was likely to be due to bleaching by the alkali treatment [11].

### 3.2. Characterization of Composite GA–NA Films with PF and PSE

The thickness of the films was increased significantly (*p* < 0.05) upon the addition of increasing levels of untreated PF or PSE, as shown in Table 6. Similar to our findings, the addition of several fillers such as alkali-treated plant fiber, bioactive plant extract, essential oils or solid nanoparticles into gum Arabic or alginate-based film matrix resulted in an increase in film thickness [26,28,34,35]. This effect is likely due to the increased solid content or variation in the film microstructure (surface roughness) by the addition of filler [26,29,35].

The water vapor permeability of films is influenced by their microstructural properties such as the porosity and presence of microcracks. The chemical interaction of the polymer matrix and the fillers with water will also play a role [11,24,35,36]. The WVP of GA–NA composite films with the addition of PF or PSE were increased compared to the control film (*p* < 0.05). There are some studies in the literature reporting the reduction in WVP with the addition of cellulosic fibers up to a certain level of incorporation, but the effect of the hydrophilic nature of the polysaccharide matrix might be more pronounced in our results [3,25,29,36]. In addition, the films with PF had significantly higher WVP values than the films with PSE at the same level of incorporation (*p* < 0.05).

The opacity of control films had the lowest value, and it had the highest transmittance among all films (*p* < 0.05). The addition of PF and PSE fillers resulted in more opaque films having less transmittance, in agreement with the literature that observed an increase in films’ opacity as a consequence of the addition of the cellulosic fibers [25,29]. The magnitude of change in opacity or transmittance was highest for films including PSE. The impermeable and insoluble nature of cellulosic fibers might be the reason for this light-hindering behavior by blocking the light passage through the composite films [25,29]. Therefore, the GA–NA composite films including PSE or PF fibers can be used for packaging of foods that are sensitive to light [25].

The color parameters of composite films were also affected by the fiber addition; the lowest brightness (L*) together with the highest greenness (−a*) and yellowness (b*) were observed in 10% PF- films (*p* < 0.05). The control film, 1% PF and 1% PSE films had similar brightness (*p* > 0.05). The reduction in L* and increase in a* and b* values were more pronounced for films with untreated fiber (PF) compared to PSE. 

The results of thermogravimetric analysis (TGA) and the derivative of mass loss (DTG) are shown in Figure 2A,B, respectively. The initial weight loss observed between 65–120 °C was related to the evaporation of free water absorbed in the films and the loss of low molecular weight compounds [11,17,28]. The second weight loss till 225 °C was rather a smooth decline for all films. The DTG (Figure 2B) was higher for the PSE film than the control and PF films, indicating that the alkali treated fibers improved the thermal stability of films compared to the neat GA–NA composite film [27,29]. The sharp decreases in the weight loss of films around 220–310 °C were attributed to the decomposition of film matrix and degradation/melting of GA, NA and glycerol [24,28,29,37]. Above that temperature range, 10% PF and 10% PSE films had almost similar decomposition behavior compared to the control film.

The FTIR spectra of the PF and PSE are given in Figure 3A. A broad peak between 3500–3000 cm^−1^ was observed for both of the fibers by stretching of O–H groups in cellulose and hemicellulose [6,17,33]. The peaks between 2960–2860 cm^−1^ of the PF and PSE had a different intensity, similar to the study evaluating the effect of alkali treatment of banyan tree roots [33], and this wavenumber range was linked with the symmetric and asymmetric C-H stretching vibration of cellulose and hemicellulose [17,29,33]. In contrast to these findings, Lim et al. [6] did not observe any intensity change with respect to the alkali treatment of bitter vine fiber. The absorption peaks at 1741 and 1773 cm^−1^ correspond to C=O stretching groups of hemicellulose and lignin [6,7]. An absorption peak at 1630 cm^−1^ can be assigned to absorbed water in the amorphous region of the fibers [6]. The bands at 1030 and 1050 cm^−1^ correspond to the C–O–C pyranose ring vibration of cellulose, C–O stretching, and C–H in plane deformation of cellulose and lignin [7,38].

The FTIR spectra of both types of the composite films are shown in Figure 3B. The wave number range of 3500–3000 cm^−1^ is attributed to the –OH groups of GA and NA, with the cellulose, hemicellulose and lignin found in fibers [24,38,39]. In addition, the incorporation of PF and PSE changed the intensity of this band range, which might be due to an intramolecular hydrogen bonding within the GA–NA matrix [29]. The spectrum of the films incorporating PF or PSE had similar bands at the same level of fiber incorporation. Although, 1% PF and 1% PSE composite films had higher intensities compared to the control and 10% fiber samples. This behavior might be related with the better distribution of fibers in the matrix at 1% level, and the compatibility of fibers with the composite matrix [29,37]. The absorption bands observed between 1180 and 920 cm^−1^ and between 900 and 700 cm^−1^ were attributed to the skeletal vibrations of the C–O bonds, ring stretching vibrations of uronic acid and out-of plane C–H bending from an aromatic structure [11,24,37].

The SEM images of the surfaces of GA–NA composite films were recorded at two magnification levels (1000× and 3000×) and are shown in Figure 4. The control film showed a more homogeneous and smooth structure without any visible microcracks or holes, which is a sign of the good compatibility and structural integrity of GA and NA [28]. However, the films incorporating fiber, especially at high levels of fiber, had rough surfaces, with a visible presence of fiber particles on the surfaces. The images of 1% PF (Figure 4(B1,B2)) and 1% PSE (Figure 4(C1,C2)) showed that the PF and PSE fibers were dispersed uniformly in the GA–NA composite matrix without obvious aggregation. There were some irregularities and microcracks in 10% PF (Figure 4(D1,D2)) and 10% PSE (Figure 4(E1,E2)) films which might be due to possible aggregation of fibers at the film surface due to high filler loading [11,26]. These findings are in line with the WVP of films, because films including the highest fiber, especially 10% PF, had the highest WVP, possibly caused by impaired film integrity due to fiber aggregation [25].

## 4. Conclusions

Parsley stalks as a food waste have the potential to be used in biobased food packaging films as filler due to the presence of high amounts of lignocellulosic fiber. The cellulose amount naturally found in the stalks is comparably lower than hemicellulose which can efficiently be removed with mild alkali treatment. This study showed that alkali extraction conditions including NaOH concentration, fiber:alkali ratio and the extraction temperature can be optimized using response surface methodology to avoid using excessive chemicals and high temperatures. Although the fiber yield, cellulose percentage, cellulose/hemicellulose, cellulose/hemicellulose and lignin percentage were selected as responses for the experimental design, the statistical results showed that only the fiber yield, cellulose % and cellulose/hemicellulose ratio can be successfully predicted with the proposed models. Optimal extraction conditions were obtained with these models and the predicted responses were found to match the experimental results. The PSE produced at the optimized conditions was incorporated into GA–NA composite films. The addition of PSE resulted in an improved WVP compared with films made with PF addition at the same level of incorporation. The optical properties of the films confirmed that these fibers can be used for packaging of light-sensitive foods as a sustainable and biodegradable non-wood cellulosic filler alternative.

## Figures and Tables

**Figure 1 foods-11-03932-f001:**
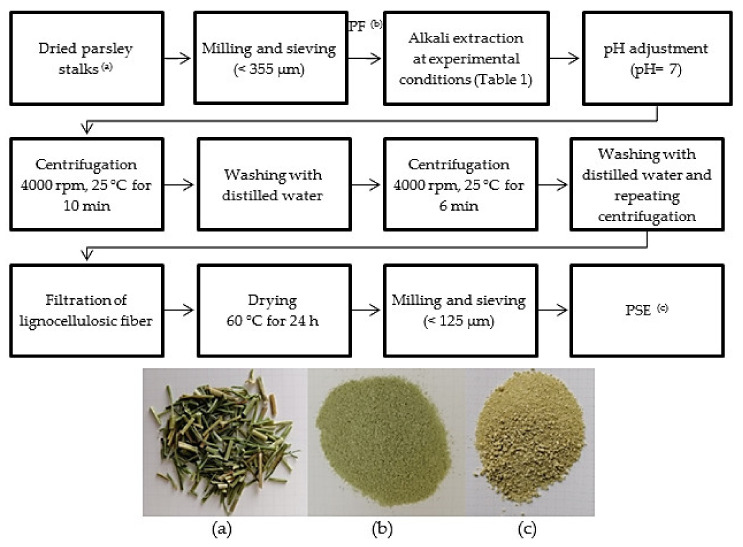
Alkali treatment steps from dried parsley stalks. (**a**) Dried parsley stalks; (**b**) PF (untreated parsley stalks fiber); (**c**) PSE (alkali-treated parsley stalks fiber extracted in run order 7).

**Figure 2 foods-11-03932-f002:**
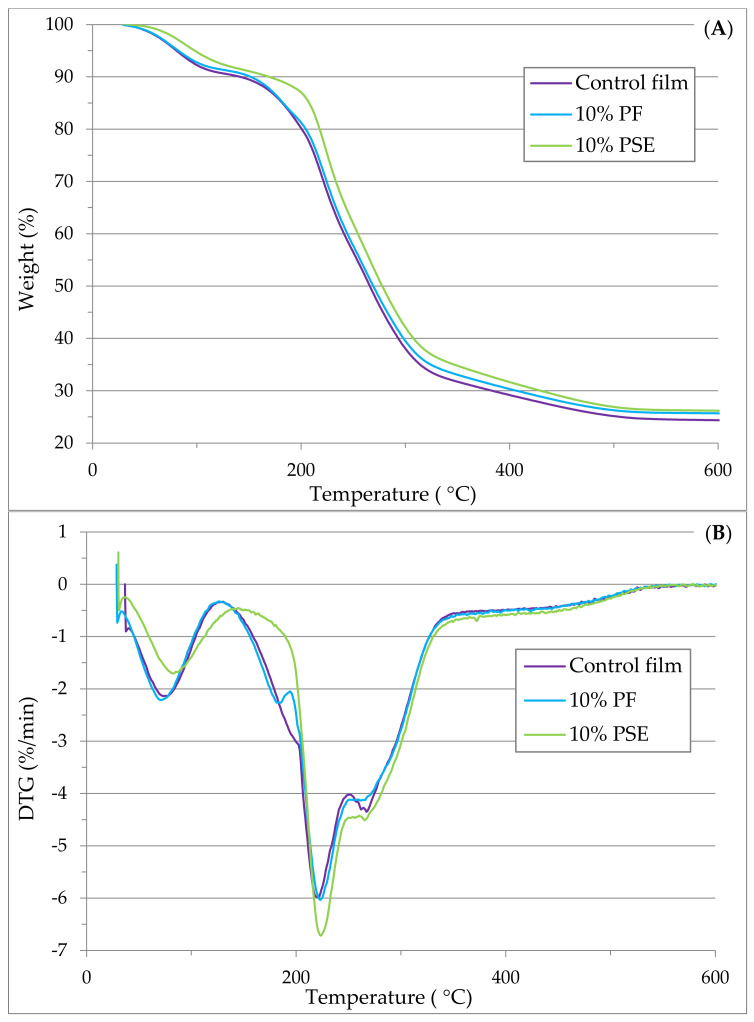
(**A**): TGA; and (**B**) DTG curves of the control film and 10% fiber containing films.

**Figure 3 foods-11-03932-f003:**
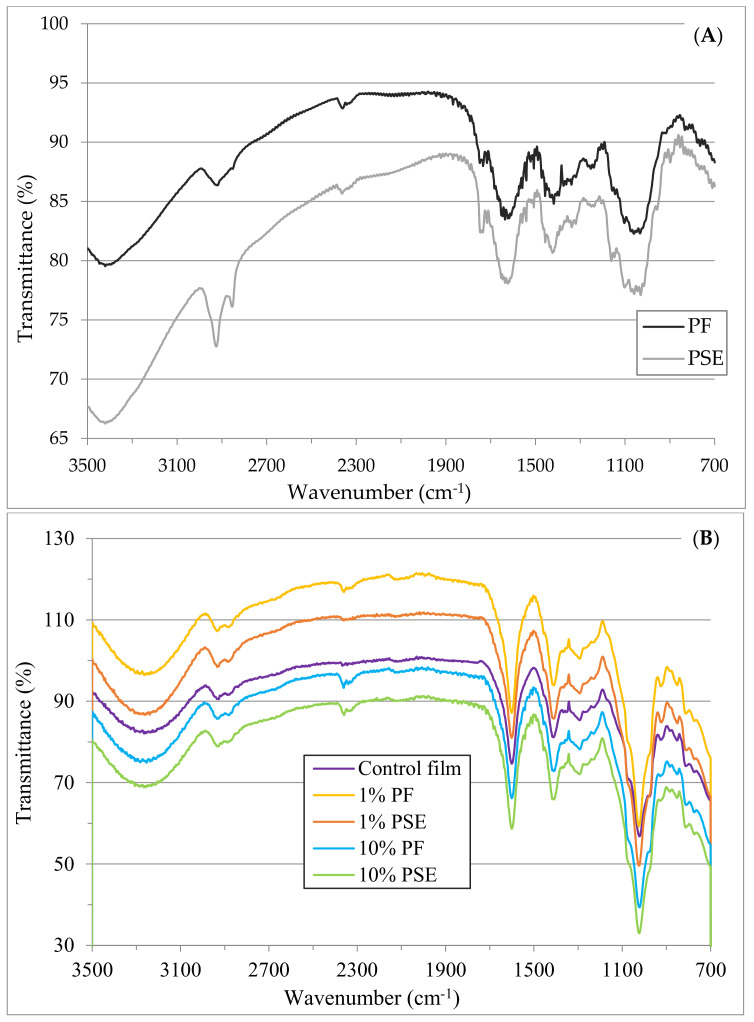
FTIR spectra of (**A**) fibers; (**B**) biobased composite films.

**Figure 4 foods-11-03932-f004:**
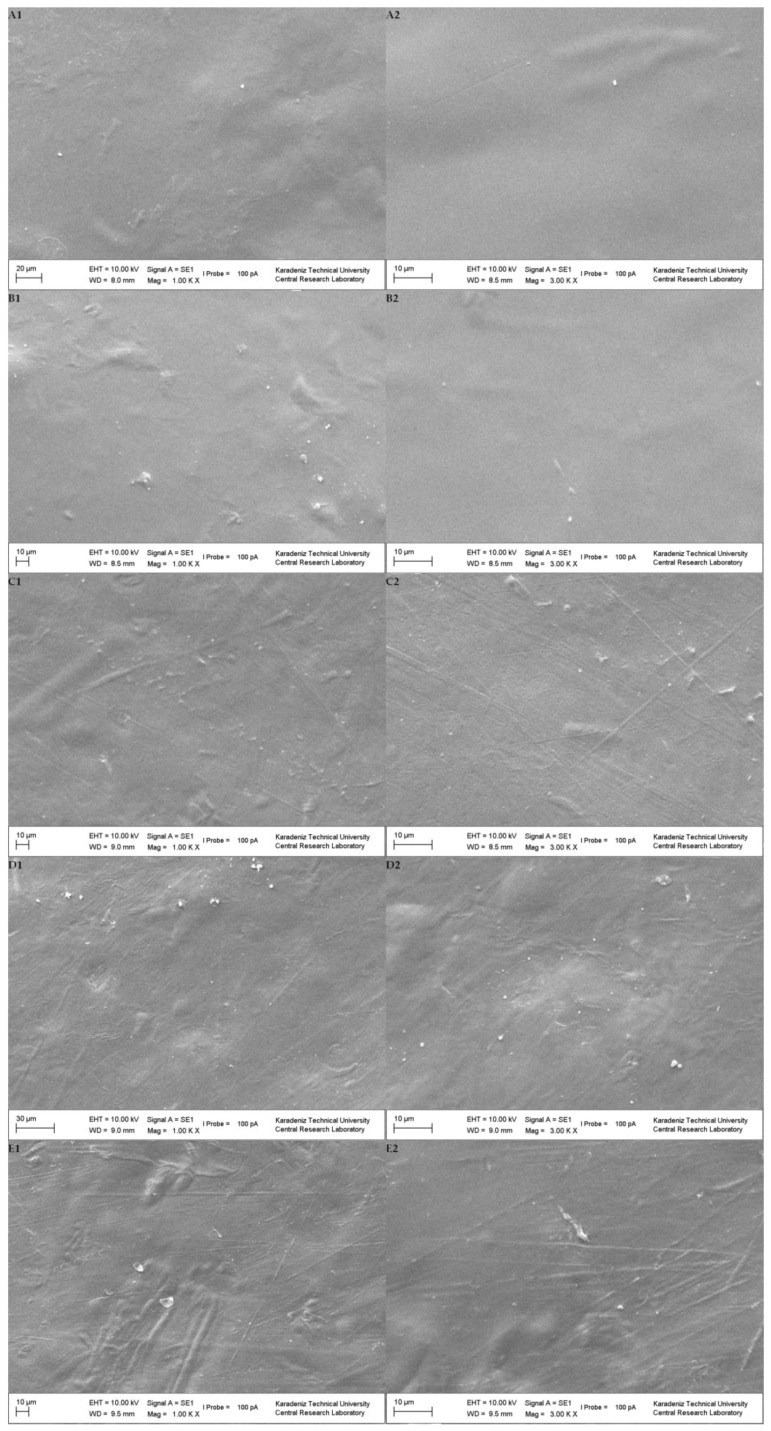
SEM micrographs, (**A1**) CF (1000×); (**A2**) CF (3000×); (**B1**) 1% PF (1000×); (**B2**) 1% PF (3000×); (**C1**) 1% PSE (1000×); (**C2**) 1% PSE (3000×); (**D1**) 10% PF (1000×); (**D2**) 10% PF (3000×); (**E1**) 10% PSE (1000×); (**E2**) 10% PSE (3000×).

**Table 1 foods-11-03932-t001:** Experimental design for alkali extraction conditions.

Run Order	Point Type ^1^	NaOH Concentration (%, *w*/*v*)	Sample/Alkali (S/A) (*w*/*v*)	Temperature (°C)
1	2	6	0.020	100
2	0	6	0.035	70
3	2	10	0.020	70
4	2	10	0.050	70
5	2	6	0.050	40
6	2	2	0.035	100
7	2	6	0.020	40
8	0	6	0.035	70
9	0	6	0.035	70
10	2	2	0.020	70
11	2	2	0.050	70
12	2	6	0.050	100
13	2	10	0.035	100
14	2	10	0.035	40
15	2	2	0.035	40

^1^ The type of point equal to zero represents center point, and two represents edge point.

**Table 2 foods-11-03932-t002:** Used factors and measured responses of the experimental design.

Run Order	NaOH (% *w*/*v*)	S/A (*w*/*v*)	Temperature (°C)	Fiber Yield (%)	Cellulose (%)	Cellulose /Hemicellulose	Cellulose /Lignin	Lignin (%)
1	6	0.020	100	25.81	35.34	0.8782	1.6139	21.90
2	6	0.035	70	30.65	19.62	0.3382	1.3244	14.81
3	10	0.020	70	32.59	22.35	0.3608	2.6827	8.33
4	10	0.050	70	29.50	32.20	0.5705	3.3049	9.74
5	6	0.050	40	46.74	20.25	0.3015	2.5171	8.05
6	2	0.035	100	34.00	35.94	0.7581	2.8772	12.49
7	6	0.020	40	43.75	17.58	0.2616	2.3061	7.62
8	6	0.035	70	33.99	20.14	0.3313	1.6144	12.47
9	6	0.035	70	34.55	22.42	0.3642	1.7714	12.66
10	2	0.020	70	43.81	23.06	0.3699	2.9787	7.75
11	2	0.050	70	44.21	19.67	0.3069	2.1625	9.09
12	6	0.050	100	23.37	44.18	1.1910	2.8002	15.78
13	10	0.035	100	19.69	55.85	2.1023	3.6480	15.31
14	10	0.035	40	41.44	21.71	0.3358	2.9288	7.41
15	2	0.035	40	49.68	27.35	0.4298	3.8714	7.07

**Table 3 foods-11-03932-t003:** Estimated regression coefficients for fiber yield (%) response.

Term *	Coefficient (Coded Value)	SE Coefficient	T	*p*
Constant	34.12	0.75	34.82	0.000
C	−6.06	0.70	−10.42	0.000
T	−9.84	0.70	−16.39	0.000
C^2^	2.74	1.02	3.36	0.020
Model				<0.0001
Lack of fit				0.6463

S = 1.6446, PRESS = 93.8211, R^2^ = 98.81%, Adj-R^2^ = 96.68%. * C: NaOH concentration (*w*/*v*, %), S: fiber sample/solvent ratio (*w*/*v*), T: temperature (°C).

**Table 4 foods-11-03932-t004:** Estimated regression coefficients for cellulose (%) response.

Term *	Coefficient (Coded Value)	SE Coefficient	T	*p*
Constant	20.02	0.87	27.48	0.000
C	3.26	0.64	7.06	0.001
S	2.25	0.64	4.86	0.005
T	10.55	0.64	22.85	0.000
C^2^	4.82	0.94	6.96	0.001
T^2^	9.84	0.94	14.35	0.000
C × S	3.31	0.90	5.07	0.004
C × T	6.39	0.90	9.78	0.000
Model				<0.0001
Lack of fit				0.4173

S = 1.3062, PRESS = 75.3716, R^2^ = 99.49%, Adj-R^2^ = 98.57%. * C: NaOH concentration (*w*/*v*, %), S: fiber sample/solvent ratio (*w*/*v*), T: temperature (°C).

**Table 5 foods-11-03932-t005:** Estimated regression coefficients for cellulose/hemicellulose response.

Term *	Coefficient (Coded Value)	SE Coefficient	T	*p*
Constant	0.38	0.07	3.21	0.024
C	0.19	0.07	2.86	0.035
T	0.45	0.07	6.84	0.001
T^2^	0.40	0.10	4.22	0.008
C × T	0.36	0.10	3.86	0.012
Model				<0.0001
Lack of fit				0.0065

S = 0.1861, PRESS = 2.7623, R^2^ = 94.91%, Adj-R^2^ = 85.76%. * C: NaOH concentration (*w*/*v*, %), S: fiber sample/solvent ratio (*w*/*v*), T: temperature (°C).

**Table 6 foods-11-03932-t006:** Film properties.

Sample	Film Thickness (µm)	WVP × 10^7^ (g/kPa.h.m)	Opacity (AU.nm/mm)	Transmittance (%)	L*	a*	b*
CF	61 ± 22 ^a^	2.244 ± 0.132 ^a^	378 ± 104 ^a^	83.46 ± 2.02 ^d^	96.42 ± 0.24 ^d^	−0.33 ± 0.03 ^f^	3.60 ± 0.30 ^a^
1% PF	82 ± 28 ^a,b^	2.838 ± 0.265 ^b–d^	396 ± 34 ^a,b^	81.55 ± 2.26 ^d^	96.33 ± 0.75 ^d^	−0.67 ± 0.10 ^e^	4.01 ± 0.79 ^a^
5% PF	91 ± 31 ^b^	3.352 ± 0.281 ^d,e^	638 ± 112 ^b^	60.30 ± 2.26 ^c^	94.54 ± 0.99 ^b,c^	−1.89 ± 0.42 ^b^	9.06 ± 2.53 ^c^
10% PF	93 ± 30 ^b^	3.592 ± 0.366 ^e^	1559 ± 288 ^d^	51.32 ± 3.32 ^b^	92.65 ± 1.30 ^a^	−2.44 ± 0.28 ^a^	15.94 ± 2.68 ^d^
1% PSE	77 ± 27 ^a,b^	2.390 ± 0.184 ^a,b^	414 ± 40 ^a,b^	81.84 ± 2.11 ^d^	96.52 ± 0.37 ^d^	−0.70 ± 0.11 ^e^	4.32 ± 0.86 ^a^
5% PSE	87 ± 21 ^b^	2.743 ± 0.230 ^a–c^	900 ± 203 ^c^	58.63 ± 3.45 ^c^	95.33 ± 0.32 ^c^	−1.14 ± 0.07 ^d^	6.84 ± 0.64 ^b^
10% PSE	93 ± 25 ^b^	3.002 ± 0.136 ^c,d^	2269 ± 217 ^e^	39.09 ± 1.90 ^a^	94.16 ± 0.28 ^b^	−1.62 ± 0.11 ^c^	9.60 ± 0.41 ^c^

CF: Control film; 1% PF: 1% untreated parsley film; 5% PF: 5% untreated parsley film; 10% PF: 10% untreated parsley film; 1% PSE: 1% optimum parsley extract film; 5% PSE: 5% optimum parsley extract film; 10% PSE: 10% optimum parsley extract film. ^a–f^ Different letters in the same column indicate statistical difference (*p* < 0.05).

## Data Availability

The data presented in this study are available on request from the corresponding author.

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
