# Peer review of "Optimization of Cellulosic Fiber Extraction from Parsley Stalks and Utilization as Filler in Composite Biobased Films"

_foods, 2022, doi:10.3390/foods11233932_

Round 1
Reviewer 1 Report
Biomass and their byproducts/wastes are rich in cellulose and could be used as the precursors to extract cellulose. Cellulose shows promising application in packaging films for foods. The research of this manuscript is interesting and results are reliable. However, revisions are required and the comments are given below.
1. “cellulose fiber” instead of “cellulose” is suggested to be added as a keyword.
2. Various biomass and their byproducts/waste could be used to extract cellulose. More references are suggested to be cited in the introduction, such as Journal of Bioresources and Bioproducts 2021, 6 (4), 323-337; Journal of Bioresources and Bioproducts 2021, 6 (2), 168-185.
3. Please pay attention to the writing of subscripts and superscripts, e.g. H2SO4 in line 74, cm-1 in line 186 need to be revised.
4. There should be a space between the number and the unit. Please go through the whole manuscript to revise it.
5. How about the effects of alkali extraction conditions on the diameter and length of as prepared cellulose fiber? SEM images or other data are suggested to be added.
6. Mechanical performance is another parameter for evaluating the films. Please add data on tensile strength, Young’s modulus, etc. Please refer and cite Journal of Materials Science 2021, 56, 9344-9355; Chinese Journal of Polymer Science 2022, 40, 764-771.
7. If the films are planned for packaging of light-sensitive foods, the antibacterial performance is suggested to be test.
8. How long can those film be degraded naturally in the soil?
Author Response
See uploaded file

Reviewer 2 Report
· Abstract: This part is very confusing, what do you mean with: alkali concentration (2-6-10%, w/v), fiber:alkali ratio (0.02-0.035-0.05; w/v) and extraction temperature (40-70-100°C). Please define the meaning of this sign (-) in the concentration, ratio, and temperature
· Optimum in what characteristics did you analysed?: The optimum conditions were determined as; 2% alkali concentration, sample:alkali ratio: 0.0276 and extraction temperature: 40°C.
· Why did you meantion the PSE cellulose fibers as filler but not reinforcing agent?
· Introduction: Please provide data about PSE lignocellulose materials abundancy, waste, etc in your region or world, if possible
· Introduction: Why did you at the preliminary stage not introduce and stress out PSE? I noted you generally describe about waste and biomass. How the correlation of PSE stalks in terms of waste? In my country PSE is still edible and used for cooking ingredients
· It is not only just mechanical properties for considering the biomass for plastic packaging, optoelectronics, etc: including food packaging, textile or paper industry due to their good mechanical properties
· This stament means that optical properties can affect mechanical nature/compatibility of the film?: improves tensile properties and the compatibility of the extracted fiber with the film matrix depending the improvements of mechanical and optical film properties.
· Confusing statement: As stated by Manna et al. [12] sample/solvent ratio ranges between 1:50 to 1:10, whereas the alkaline concentration ranges generally between 0.8-15% although up to 40% was found in the literature [10, 11, 13,14].
· Introduction: There is no information about PSE, such as chemical composition, and why the PSE is beneficial for plastic and so forth?
· In the material, you mentioned the use of H2SO4 but there was no use of the chemical in the procedure of PSA cellulose isolation
· Why did you not use freeze drying but oven drying after obtained PSE cellulose? It can cause aggregation but then you milled again to obatained finer cellulose
· What is about extractive content? Cellulose amount (%) was calculated from subtracting H%, L% and ash % from 100.
· No aggretaions of PSE celluloe? Please check this statement: fibers smaller than 125 µm were added into the film forming solutions at 1-5 143 and 10% (w/w, on GA+NA basis) after the mixing with glycerol.
· Why did you divide the absorbance area with film tchiness to measure opacity?
· If so, NaOH used for delignification did not work? two-fold reduction of hemicellulose which was the main component in the lignocellulosic structure of PF
· Table 2. Please provide the percentage of ash
· Formula 3, 4, and 5: What are the implications of these formulas?
· Calculation of extractives need to be performed so that the regressions 3, 4 and 5 can be used.
· Table 4, please check again the units of tested parameters
· Figure 2: TGA profiles, please make the line of being smooth and thin not bold. Please make the figure clearer
· Figure 3 is not clear and overlapped. Please make distictions
· Why SEM Figure for B2 is smoother than B1?
· There is no conclusion in the manuscript. Please make it
Thank you so much

Author Response
See uploaded file

Reviewer 3 Report
Congratulation to the authors for a very good paper. Just add full names of abbreviations in line 58. To my opinion lines in fig 3 could be thinner so the picture would be clearer.
The paper addresses the possibility of the use of lignocellulosic fiber from parsley stalks in gum Arabic-sodium alginate composite films for improving the characteristics of the biobased film. Production and application of biodegradable films is one of the most important and developing topics in research in the last two or three years. So, improvement of the films characteristics has scientific soundness and interest to readers, especially as it gives a possibility of packaging of light sensitive commodities. The authors did a great job in writing the papes since it is very clear and easy to read and understand. The conclusion is supported by the results and confirms the aim of the paper.
Author Response
See uploaded file

Round 2
Reviewer 1 Report
Minor revisions are still required for this manuscript. The comments are listed below.
1. There should be a space between the number and the unit. “100°C” and other units need to be revised.
2. Biomass derived materials are promising for food packaging. Many interesting researches have been done on this field. Some typical references are suggested to be added in the introduction for broad readers, e.g. Journal of Bioresources and Bioproducts 2022, 7 (1), 1-13; Journal of Bioresources and Bioproducts 2021, 6 (2), 168-185.
3. “ml/ min” and “oC/ min”, etc. are suggested to be revised as “ml/min” and “oC/min”, respectively.
Author Response
Thank you for your comments
We changed the units according to your suggestion.
We already included several papers on using bioresources for packaging.
Reviewer 2 Report
Dear Editor and Authors,
Good morning and thank you for entrusting me in reviewing this manuscript.
After reading the last version of the manuscript. I found TGA and DSC profiles of the manuscript are still unreadable. Please don't make a very close overlapping profiles, and I would like to recommend you to revise again the profiles by:
1. Please just add about 10-20 transmittance increase of each samples tested for TGA/DSC test so that it can make your thermal stability profiles closely overlapped
2. If could not, please just make the line of the bands in the profiles not thick but thin with colour lines.
Thank you so much and I am looking forward your revision
Best regards
AS
Author Response
Thank you for your suggestions, we improved the Figures 2 and 3 to make the lines more distinguishable